# All-Polymeric Electrode Based on PEDOT:PSS for In Vivo Neural Recording

**DOI:** 10.3390/bios12100853

**Published:** 2022-10-10

**Authors:** Gilberto Filho, Cláudio Júnior, Bruno Spinelli, Igor Damasceno, Felipe Fiuza, Edgard Morya

**Affiliations:** 1Edmond and Lily Safra International Institute of Neuroscience (ELS-IIN), Macaíba 59280-000, Brazil; 2Department of Materials Engineering, Federal University of Rio Grande do Norte (UFRN), Natal 59072-970, Brazil

**Keywords:** PEDOT:PSS, neural recording, immune response, BMI

## Abstract

One of the significant challenges today in the brain–machine interfaces that use invasive methods is the stability of the chronic record. In recent years, polymer-based electrodes have gained notoriety for achieving mechanical strength values close to that of brain tissue, promoting a lower immune response to the implant. In this work, we fabricated fully polymeric electrodes based on PEDOT:PSS for neural recording in Wistar rats. We characterized the electrical properties and both in vitro and in vivo functionality of the electrodes. Additionally, we employed histological processing and microscopical visualization to evaluate the tecidual immune response at 7, 14, and 21 days post-implant. Electrodes with 400-micrometer channels showed a 12 dB signal-to-noise ratio. Local field potentials were characterized under two conditions: anesthetized and free-moving. There was a proliferation of microglia at the tissue–electrode interface in the early days, though there was a decrease after 14 days. Astrocytes also migrated to the interface, but there was not continuous recruitment of these cells in the tissue; there was inflammatory stability by day 21. The signal was not affected by this inflammatory action, demonstrating that fully polymeric electrodes can be an alternative means to prolong the valuable time of neural recordings.

## 1. Introduction

One of the significant challenges today in the brain–machine interface using invasive methods is the stability of chronic recording. Invasive microelectrodes are generally used to microstimulate or record brain activity close to neurons [1]. Currently, extracellular recording methods make it possible to obtain two different types of signals—the local field potential (LFP), which reflects the current of multiple neurons, and the action potential, which reflects the firing of only one neuron [2]. However, the immune response to the implant, which is a foreign body, is considered a primary source of instability and signal loss. This is caused by the proliferation and activation of glial cells, specifically astrocytes, and microglia, at the implant site. Glial-mediated inflammatory processes might generate a glia scar that encapsulates the tip of the microelectrode dampening signal recording and electrical stimulation. Ultimately, microelectrodes might become unusable after this process [3,4,5].

In terms of the materials used, two mechanical properties are considered key—Young’s Modulus and flexural stiffness [6]. Although the Young’s modulus defines the intrinsic properties of a material, flexural stiffness plays a more direct role in determining mechanical incompatibility, and therefore, in the immune response provoked in brain tissue [7]. The materials used in conventional recording electrodes, such as metal, carbon, and silicon, are approximately 8–9 orders of magnitude stiffer than the brain tissue inserted into them. Thus, there is a need to develop more flexible electrodes that try to emulate the properties of the tissue where they will be implanted, not only considering the mechanical properties, but also the relation. Polymer-based electrodes are gaining more and more emphasis in research due to an approximation of the brain tissue property, as presented in [8,9,10,11,12].

Aiming to increase the ability to perform the neural recording for a longer time and in an attempt to approximate the mechanical resistance of the materials used with that of brain tissue, we carried out the manufacturing of flexible microelectrodes based on PEDOT:PSS and molds build with 3D printing, in an attempt to cause minor damage to the tissue where it will be implanted, and in this way, can record electrophysiological signals from the cerebral cortex for a longer time due to a possible reduced immune response to the invading material.

## 2. Materials and Methods

### 2.1. Manufacturing of All-Polymeric Electrodes

For the fabrication of the electrodes, 3D-printed molds (S3, Sethi3D, Campinas, Brazil) were designed using CAD software (Fusion 360, Autodesk, San Rafael, CA, USA) for intracortical electrodes. The molds were printed with a 0.4 mm nozzle and 1.75 mm ABS filament. Polydimethylsiloxane (PDMS) was obtained by mixing it (95:5 %wt) with a catalyst. Then, the PDMS was deposited on the negative mold and left for 24h in an oven (Bio SEA 150L, 7Lab, Rio de Janeiro, Brazil) at 37 ${}^{\circ }$C. After 24 h, the PDMS was manually removed from the negative mold. PEDOT:PSS-based ink (PH 1000, Clevios, Heraeus, Hanau, Germany) was obtained through the steps previously described by [13], where PEDOT:PSS was stirred for 6 h at room temperature and filtered through a syringe filter (0.22 μm). The filtered PEDOT:PSS solution was frozen by immersion in liquid nitrogen. The frozen PEDOT:PSS was then placed in a lyophilizer (L101, Liobras, São Carlos, Brazil) and lyophilized for 72 h. PEDOT:PSS nanofibrils were redispersed with a mixture (85:15 *v*/*v*) of deionized water and dimethyl sulfoxide (DMSO) (Sigma-Aldrich, San Luis, USA), followed by mechanical agitation for homogenization. The deposition of the PEDOT:PSS ink was performed manually through a syringe on the PDMS negative mold (400 μm thick and 200 μm high for deposition), where after deposition, it was left to dry in an oven at 35 ${}^{\circ }$C. After that, the electrode tracks were isolated with PDMS (400 μm thick). After isolation, the electrode was placed on an acetate plate to connect the PEDOT:PSS lines with silver ink to the Omnitics direct channels (32 channels, Omnetics 1125, Omnetics Connector Corporation, Minneapolis, USA). This method manufactured for intracortical recording has four channels (400 μm in diameter each) for electrophysiological recording.

### 2.2. Conductivity Measurement

The electrical conductivity measurement of the PEDOT:PSS ink was performed using the four-probe method according to ASTM-F43-99 [14]. The conductivity measurement was based on the method previously described [13]. The equipment was used for the point probe, the contact tips were placed with a spacing of 2 mm, and the following formula of conductivity (σ) was applied:(1)$$\sigma =\frac{I\times D}{V\times L\times E},$$
where *I* is the current applied to the sample, *D* is the distance between the two electrodes that will measure the voltage, *V* is the voltage between the electrodes, *L* is the width of the sample, and *E* is the thickness.

### 2.3. Impedance Measurement

The impedance measurement of the electrode channels was performed using an impedance meter (EIT-201, Thomas Recording, GmbH, Giessen, Germany) at 1 kHz, where the channels were immersed in saline solution (0.9%), and a reference electrode and grounding electrode were used in the system. Measurements were performed with signal stabilization after 15 min of connecting the measurement system.

### 2.4. Scanning Electron Microscopy

Images were obtained using a scanning electron microscope (SEM) (Auriga 40, Zeiss, Jena, Germany), and further analysis was performed using ImageJ software (ImageJ 1.53k, National Institute of Health, Bethesda, MD, USA).

### 2.5. In Vitro Recording

The fully polymeric electrode was used to record electrophysiological signals in vitro in saline solution (0.9%), and the cable was connected to a preamplifier (Plexon Inc., Dallas, TX, USA) connected to the Omniplex® data acquisition equipment (Plexon Inc., Dallas, TX, USA). The stimulus was placed in saline solution with the electrode and recorded with an acquisition rate of 1000 Hz. The artificially generated signal was also recorded with the Headstage Tester Unit (HTU) (Omniplex, Plexon Inc., Dallas, TX, USA) with an acquisition rate of 1000 Hz.

The signal-to-noise (SNR) was used to validate the signal quality obtained by the electrode. SNR is defined as the ratio of the power spectral density (PSD) of the signal to the power of the noise. In this way, we recorded the artificial signal by the HTU and subtracted it from the measured signal from the electrode to find our noise. The SNR can be obtained by: (2)$$SNR_{dB}=10\mathrm{lg}10\left(\frac{P_{signal}}{P_{noise}}\right),$$
where $P_{signal}$ is the mean power of the signal and $P_{noise}$ is the mean power of the noise.

### 2.6. Animal Handling

All experiments described were accepted by the Ethics Committee on the Use of Animals (CEUA), from the Edmond and Lily Safra International Institute of Neurosciences, according to the guidelines of the National Council for Animal Experimentation (CONCEA), under the protocol 01/2022. The animal model used in this study was Wistar rats belonging to the vivarium of the IIN-ELS, which were housed in housing boxes (up to a maximum of three animals per box) in the period prior to the procedure. However, after electrode implantation, the animals were kept in insulated boxes with light cycles, adequately monitored, and had free access to water and food.

### 2.7. Surgical Implantation

Wistar rats were anesthetized with isoflurane inhalation (induction of 0.8–1.5 lmin${}^{-1}$ oxygen flow, 4–5% isoflurane, state maintenance with 0.8–1.5 lmin${}^{-1}$ oxygen flow, and 2–3% isoflurane). Anesthesia was performed with intramuscular application of atropine (0.05 mg/kg), xylazine (3 mg/kg), and intraperitoneal application of ketamine (70 mg/kg). To confirm that the animal was anesthetized, the paw reflex was evaluated. Gazes moistened with saline (0.9%) were placed over the eyes to prevent them from drying. The animal’s temperature was maintained at 37 ${}^{\circ }$C using a heated bed during the entire surgical procedure. The animal’s head was shaved to expose the skin, followed by subsequent positioning of the stereotaxic and making a cut to access the skull, which was properly sanitized before marking the coordinates with the stereotaxic. Trepanations were performed in the coordinates (1.0:2.5 anteroposterior (AP), 2.8:−4.2 mediolateral (ML); and for the implantation of the electrode, 0:2.5 dorsoventral (DV)) to perform the craniotomy. Then, the dura was removed from the window with tweezers. In the next step, the screws were fixed to the skull, followed by the sequential descent of the fully polymeric electrode array inside the open windows. Finally, an acrylic helmet was made around the electrodes and screws to protect these elements.

### 2.8. Neural Recordings

Rats were initially sedated with isoflurane inhalation (induction of 0.8–1.5 lmin${}^{-1}$ oxygen flow, 4–5%), with subsequent headstage docking. Sequentially, the animals were placed in an acrylic box for recording free behavior, and the cable was connected to a preamplifier (Plexon Inc., Dallas, USA) connected to the Omniplex® data acquisition equipment (Plexon Inc., Dallas, USA). Field potentials (LFP) were recorded with an acquisition rate of 1000 Hz, with the rat being under the influence of isoflurane for approximately 15 min. After the effect of isoflurane, the field potentials (LFP) of the rat in free movement in the box were recorded for 15 min with an acquisition rate of 1000 Hz. After the recordings, a bandpass filter from 0.5 to 40 Hz was used, and the LFP of each channel was analyzed. The power density spectrum was calculated for each record, both for the anesthetized animals and for the animals in a free movement. Data were processed and analyzed using the Python language.

### 2.9. Euthanasia

The animals were anesthetized and euthanized using the exsanguination technique through intracardiac perfusion using 4% paraformaldehyde (4% PFA; pH 7.2) and with the aid of an extracorporeal perfusion pump. The brain was therefore removed from the skull and placed in a solution containing 4% paraformaldehyde (4% PFA; pH 7.2) and kept at 4 ${}^{\circ }$C for 24 h.

### 2.10. Histological Analysis

Brain samples were placed in a 30% sucrose solution for 3–4 days. The brains were then covered with Tissue Tek^®^, frozen in a cryostat (HM550, Thermo Fisher Scientific, Waltham, MA, USA) (cabin and specimen temperature −20 ${}^{\circ }$C), and sectioned into 50 μm coronal slices. The sections were stored in cryoprotectant solution (50% glycerol in PBS), and those considered of interest were washed three times in PBS for 5 min with minimal agitation. The tissues were then incubated in 1% PBST (99% PBS + 1% Triton™X100) at room temperature and minimal agitation for 1 h, and then incubated for a further 1 h in a 10% goat serum solution (NGS) in 0.3% PBST. Incubation with primary antibodies (Ab) was performed for 72 h at 4 ${}^{\circ }$C for primary antibodies to glial fibrillary acidic protein (GFAP obtained in rabbit; 1:500; Z0334, Dako, Agilent, Santa Clara, CA., USA) and cluster of differentiation 68 (CD68 obtained in mouse, 1:500; MCA341R, AbD Serotec, Bio-Rad Laboratories, Inc., Hercules, CA, USA). Therefore, the sections were washed in 0.3% PBST and incubated for 2 h 30 min with the secondary antibodies (Mouse-Alexa Fluor 555 nm, Thermo Fisher Scientific, Waltham, MA, USA, 1:1000; brand and Rabbit-Alexa Fluor 488 nm, Thermo Fisher Scientific, Waltham, MA, USA, 1:1000;) in 10% NGS + 90% 0.3% PBST at room temperature with minimal agitation. Sequentially, sections were washed in PBS for 5 min three times and subsequently incubated with DAPI (1:1000) in PBS for 10 min. Finally, samples were washed with PBS for 5 min, five times, and mounted. The slides were stored at 4 ${}^{\circ }$C until the images were acquired.

### 2.11. Image Acquisition and Analysis

Multichannel images were obtained using a fluorescence microscope (Imager.Z2, Zeiss, Jena, Germany) (20× objective; UV, FITC, and Texas-red fluorescent filter cubes) using an 8-bit RGB scale with the aid of Stereo Investigator software. The images were analyzed in Python, using the scikit-image library.

## 3. Results and Discussion

### 3.1. Fabrication and Characterization of the All-Polymer Electrode

The use of 3D printing to manufacture electrodes has become an interesting alternative for the area due to its easy customization and low cost [13]. In this sense, molds were made for the manufacturing of electrodes from these negative intracortical molds. PDMS was used as a substrate and insulator in the electrodes due to its ability to be inert (causes a lower immunological reaction) and excellent biocompatibility. PEDOT:PSS, in turn, is commercialized in an aqueous form and has been widely used in several areas. In addition, it is commonly used with elastomers to manufacture cardiac and neuronal sensors [15,16], and in this sense, it was also chosen as a conductive material in this work due to its high degree of biocompatibility. PEDOT:PSS was doped with DMSO to increase its electrical conductivity [17]. Modifications of PEDOT:PSS viscosity are described as one of the ways to produce PEDOT:PSS ink for 3D printing [13].

The PEDOT:PSS:DMSO ink was manually deposited on the PDMS substrate (400 μm thick), whose contacts were insulated with PDMS layers (400 μm) (Figure 1a), in which each contact channel was 400 μm in diameter (Figure 1b). In the model, the layer height and the adhesion of the PEDOT:PSS:DMSO deposition on the PDMS substrate were analyzed using scanning electron microscopy (SEM), since PDMS commonly has hydrophobic characteristics [18]. A layer height of 10.5 ± 1.2 μm was found in the channels deposited in the intracortical model (Figure 1c). The adhesion obtained on the electrodes was considered satisfactory, demonstrating the ability of the PEDOT:PSS:DMSO ink to adhere to the PDMS surface without any surface treatment, consequently making the electrode fabrication less complex. The electrical conductivity of PEDOT:PSS varies according to the method of obtaining and doping methods. Some studies show that PEDOT:PSS doped with DMSO can reach values of up to 1500 S/cm [19], but when removed from the water and 3D printed, the values reach 155 S/cm [13]. The values obtained through the measurement by the four-point method were 137 S/cm for the PEDOT:PSS:DMSO ink. This result is in agreement with other findings in the literature [13,20,21], in which a nozzle of 400 μm in diameter was used according to the manufacturing method used, and consequently, the non-orientation of the PEDOT nanofibrils was observed: PSS, thereby causing a decrease in conductivity. The electrical resistance values of manually deposited PEDOT:PSS:DMSO were analyzed, and values of 180.7 ± 19.5 Ω were observed. In this context, it is noteworthy that impedance is one of the fundamental characteristics of an electrode used for neural recording, and its respective values tend to directly influence the signal-to-noise ratio of the electrode [22]. Impedance values at 1 kHz of 7.4 kΩ were obtained for the electrode with a diameter of 400 μm, which demonstrates the excellent applicability of PEDOT:PSS:DMSO as a conductor for neural recording. Other studies showed electrode impedance values using PEDOT:PSS as a conductor that ranged from up to 6.3 kΩ [23] to 19.5 kΩ [24].

The fully polymeric electrode had four 400 μm channels for contact with brain tissue, in which each channel was connected to the omnetics via silver ink lines and insulated with a light-curing resin. Sequentially, at the end of the procedures, the electrodes were weighed on a precision balance, and we obtained weight of 0.5 g per prepared flexible electrode. It was considered that the weight of the complete electrode was within the expected range, since manufacturing the fully polymeric electrode was done manually.

### 3.2. Validation In Vitro

The fully polymeric electrodes were tested with artificial signals to analyze the recording capacity at an acquisition frequency of 1000 Hz. In this sense, all electrode channels demonstrated excellent ability to record the generated signals and stability over the analysis time. We compared the signal recorded by the electrode with the signal recorded by the Headstage Tester Unit (HTU) equipment, which tests the quality of the signal acquired by the headstage coupled to the plexon system. It was found that both received the same signal and the same acquisition frequency. A bandpass filter (0.5–35 Hz) and the calculation of LFP (Figure 2a,b) and PSD (Figure 2c) were used to compare whether the signal frequencies were the same. It was possible to verify a difference in the form of the signal recorded by the flexible electrode compared with the signal from the HTU, and in this context, the characteristics of the material and the form of recording of the electrode are different from those of the HTU. In the HTU, the audio cable is connected directly to the board, and in the test with the electrode, it was submerged in saline solution (0.9%) and had a wire transmitting the signal to the solution. From the HTU signal, it is possible to calculate the SNR (Figure 3), for which we used the difference between the signals of the fully polymeric electrode with the HTU and thus found an SNR ratio of 12dB, which demonstrates a good signal-to-noise ratio. Studies show that the maximum SNR value at low frequencies (5–30Hz) is approximately 9dB for Pt and CNTs, and 4dB for Au [25]. This SNR value demonstrates that the electrode based on PEDOT:PSS:DMSO has interesting applicability for recording low-frequency LFPs.

### 3.3. Validation In Vivo

To evaluate the intracortical model of the fully polymeric electrode based on PEDOT:PSS:DMSO, neural recordings were performed in Wistar rats. The electrodes were implanted (Figure 4), focusing on the region of interest of the primary motor cortex (M1), and neuronal activity was recorded at two different times: first, the recording was in an acrylic box with the animal still under the effect of isoflurane anesthesia for 15 min. In this state, the animal remained anesthetized and immobile until the anesthetic’s effect wore off, so it returned to the awake state. In the second moment, the neuronal recording was performed with the animal having free movement inside the acrylic box. LFPs were recorded at rest at two moments: anesthesia and free movement. To test the ability of long-time recording, animals were recorded 7, 14, and 21 days after implantation to compare the functionality of the channels as a response to the implant progression. The data were filtered with a bandpass filter (0.5–35 Hz), as the frequencies of interest for recording are contained in this range. We found differences between the LFPs in the conditions: we observed that delta frequencies (0.5–4Hz) are more present when the animal is anesthetized compared with free movement (Figure 5a), which is in agreement with studies that analyzed the influence of anesthesia on the frequency of neuronal activity [26]. The PSD between states was also calculated, demonstrating the greater intensity of delta waves in the anesthetized animal compared to the moving animal (Figure 5b).

The analysis of the animal in the free-movement condition was performed. The spectrogram of the mouse during movement showed changes in frequencies for when it performed movements (Figure 6). Therefore, it was cut to just 1 s of motion, and the PSD and spectrogram were calculated (Figure 7). The frequencies that proved to be predominant were those expected for the M1 region when performing the movement, which are alpha frequencies (8–12 Hz) [27].

The functionality of the electrode channels over the three weeks was verified with the raw and filtered LFP, and the signals were stable after 21 days of surgery and with all functional channels after this period Figure 8. This result is essential for lead utility issues, as previous studies showed that for PEDOT:PSS-based fully polymeric leads, only 62% of the channels remained functional for two weeks and 45% for four weeks [28].

### 3.4. Progression of the Glial Response

The spatiotemporal progression of glial responses to the intracortical electrode implant was characterized. Electrodes were operated with the same characteristics and in the same region. The immune response evolution was observed 7, 14, and 21 days after implantation of the fully polymeric electrodes based on PEDOT:PSS:DMSO.

For each rat, brain tissue was sectioned in the coronal plane. The images of the regions which interfaced with the electrode were obtained with different magnifications and were later analyzed using the scikit-image library. The images had their pixel intensities normalized; that is, the average vector of the intensity of each image was obtained, and values below the average were zeroed. Then, the interface between the electrode and the tissue was analyzed by lining up a square (1000 μm × 1000 μm) on each side where the electrode was implanted in the tissue, that is, the left and right interfaces of the electrodes. Means were calculated for each time and interface (right and left), along with the intensity of each marker, by the distance from the implant site. Images for each time-point and CD68, GFAP, and DAPI stains were performed to evaluate glial response progression (Figure 9).

**Figure 8 biosensors-12-00853-f008:**
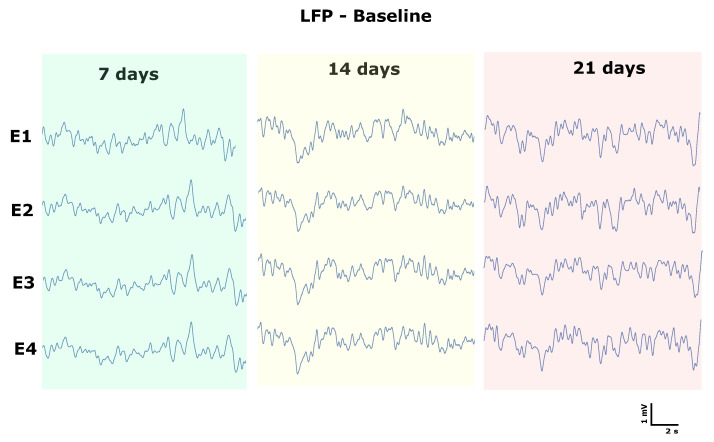
Analysis of functioning and intensity through the LFP of days 7, 14, and 21 of all channels.

The intensity of each signal was analyzed in different periods, for which the average vector of the intensity was divided by the maximum value in this analyzed square (Figure 10a,b) and the intensity was verified by the distance of the electrode interface from the tissue (Figure 10c). CD68 is a type of marker-activated microglial cell in the nervous system; that is, when there is a proliferation of this type of cell, there will be an increase in the signal intensity of this marker [29]. In this sense, the intensity of the CD68 marker was very apparent in the first 7 days (Figure 10d), and it was possible to verify migration of these cells to the interface with the electrode after 14 (Figure 10e) and 21 days (Figure 10f). The GFAP marker, which was used to label astrocytes, was also used, and there was activation of these cells, with migration of the electrode interface after 7 (Figure 11d), 14 (Figure 11e), and 21 (Figure 11f) days.

The average intensity of each image vector relative to the signals was calculated in the selected quadrant and divided by its maximum value. In this way, a graph was obtained in which the intensities of these vectors are related to their distances from the tissue–electrode interface for both sides. It is possible to notice that the CD68 intensity was higher in the first seven days after implantation for a greater distance from the interface, which demonstrates that the recruitment of microglia to the region affected by the electrode lesion occurs early (Figure 10d), as shown by other studies [30]. After 14 days, it was possible to verify a higher concentration of microglia at the electrode interface (Figure 10e), and the characteristic of migration to the interface remained present (Figure 10f). The GFAP signal intensity tended to increase over time. In the first 7 days, there was not much signal intensity close to the lesion, but a greater distribution was found further away from the site, including considerable values at 500 μm from the damage (Figure 11d). After 14 days, it was also possible to verify increases in the signal intensity and in the recruitment of astrocytes with migration to the electrode–tissue interface (Figure 11e). The intensity of the GFAP signal tends to increase close to the lesion site, and the distance of this intensity decreases as it moves away from the interface, showing that the recruitment of astrocytes decreases considerably (Figure 11f).

**Figure 11 biosensors-12-00853-f011:**
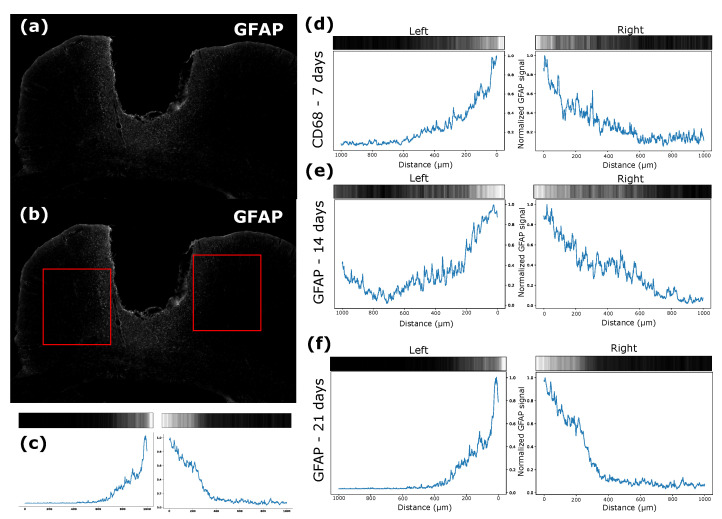
(**a**) Image after treatment with averaging of the intensity vectors. (**b**) Quadrant used to measure GFAP intensity. (**c**) Signal strength near the electrode–tissue interface. (**d**) GFAP signal strength near the electrode–tissue interface after 7 days. (**e**) GFAP signal strength near the electrode–tissue interface after 14 days. (**f**) GFAP signal strength near the electrode–tissue interface after 21 days.

Intensity normalization was performed every day, making the largest value of the vector between the periods become the value at which it would be compared between all. The intensity and distance from the electrode–tissue interface of CD68 tended to be greater (approximately 400 μm away) in the initial seven days; as the days progressed, the intensity tended to decrease, and the proliferation got closer to the interface (100 μm) by days 14 and 21, the latter having the lowest intensity of the three (Figure 12). Our results resemble those of a previous study, in which the implant of a silicon intracortical electrode generated increased CD68 and GFAP staining around the tip [31]. After 2 weeks post-implant, GFAP intensity was higher than the CD68 signal, and both were greatly decreased after 4 weeks [31]. Moreover, in other previous study, a tissue–electrode was dissected for evaluation of microglial and astrocyte-associated gene expression [32]. The outcomes suggest that genes related to microglial activation and inflammation start up-regulation in the early days post-implant (24 h to 1 week). On the other hand, genes related to astrocyte differentiation are overexpressed over a longer time-frame, starting after 1 week [32]. These spatiotemporal changes are parts of the role of microglial cells in producing quick inflammatory responses, such as secretion of proinflammatory cytokines (e.g., IL-1 IL-6, and TNFα), which diminishes when the injury site is normalized [30]. Astrocytes, however, produce a longer morphological response, aiming to isolate the injured cellular environment [30]. In this way, the projection of these microglial cells in the vicinity of the electrode is expected, and they tend to stay at the interface. On the other hand, the intensity of GFAP was the opposite to that presented by cells marked by CD68, since GFAP marks astrocytes. GFAP intensity was lower in the first 7 days, but with recruitment further away from the lesion. As the days went by, the signal strength tended to increase (see days 14 and 21), but the distance to the signal interface decreased (Figure 12). When tissue damage occurs, astrocytes become reactive, proliferate, and undergo hypertrophy [3]. These have a slower response to damage, but there is migration to the electrode interface [30]. This phenomenon is known as glial scarring, in which the electrode surface undergoes encapsulation and its recording capacity is compromised [30,33].

## 4. Conclusions

The present work demonstrated the manufacturing of fully polymeric electrodes based on PEDOT:PSS:DMSO. We also showed biological evidence that these electrodes are a biocompatible alternative for signal acquisition in in vivo neural recordings. The conductive ink based on PEDOT:PSS:DMSO obtained 137 S/cm of electrical conductivity and 180.7 ± 19.5 Ω electrical resistance. Electrodes were obtained with four channels of 400 μm in diameter, with an impedance of 7.4 kΩ. The validation of the electrode in vitro demonstrated an SNR of 12 dB compared to the HTU, with an attractive signal-to-noise ratio compared to some materials used in electrodes for recording. Neural activity records were made of Wistar rats in two different conditions to validate the electrode; it was possible to analyze LFPs in these conditions, demonstrating the ability to perform such a function even in rats with free movement. The immunological response to the electrode was measured using immunohistochemistry with specific markers for cells belonging to the nervous system—CD68, and GFAP for microglia and astrocytes, respectively. We demonstrated that the fully polymeric electrode has an interesting immune response, which is completed around 21 days after implantation, without showing any significant change in the quality of the signal recorded by it.

In general, the fully polymeric electrode based on PEDOT:PSS:DMSO proved to be an effective way to perform neural recording in vivo, demonstrating an adequate signal-to-noise ratio and good stability over the period studied. The immune response showed that glial encapsulation occurs up to 21 days after implantation without showing any damage to the electrode channels. The next step towards understanding long-term recording properties would be to use a more extended period of time, that is, recording for a while more than six months, with recording intervals of one or two weeks, in addition to reducing, even more, the size of the electrode. The excellent stability of the device–tissue interface showed to be a promising way to use fully polymeric electrodes in the neural recording.

## Figures and Tables

**Figure 1 biosensors-12-00853-f001:**
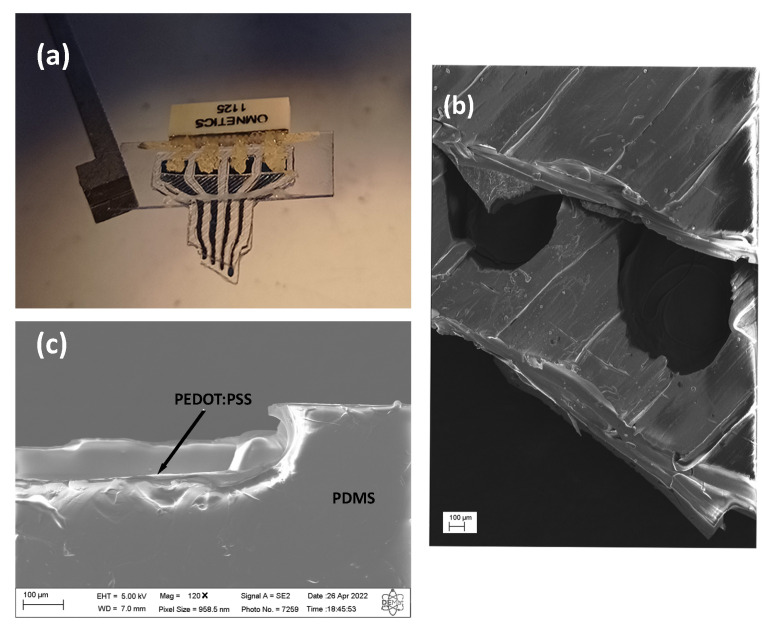
(**a**) Intracortical electrode. (**b**) Image of the intracortical channel obtained with SEM. (**c**) Interface deposition with PEDOT:PSS:DMSO and PDMS, layer deposition of 10.5 ± 1.2 μm.

**Figure 2 biosensors-12-00853-f002:**
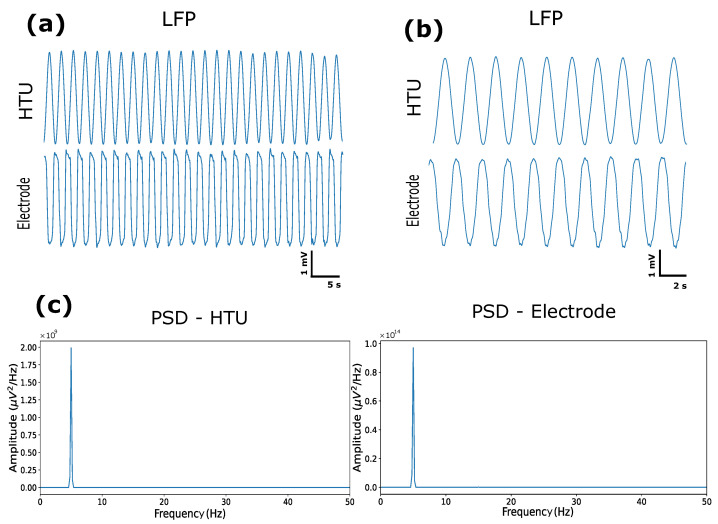
(**a**) Artificial signal LFP with 5 s for the HTU and electrode. (**b**) Artificial signal LFP with 2 s for the HTU and electrode. (**c**) Artificial signal PSD with the HTU and electrode.

**Figure 3 biosensors-12-00853-f003:**
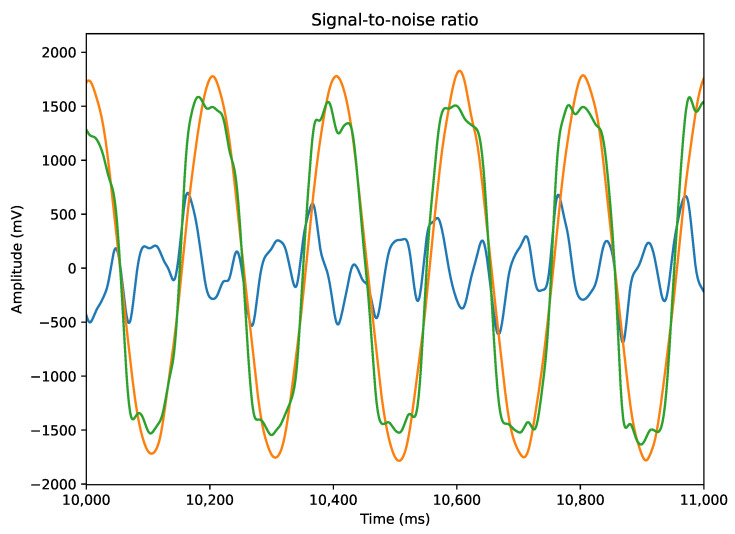
Difference between the signals detected by the HTU and by the electrode, and the influence of noise.

**Figure 4 biosensors-12-00853-f004:**
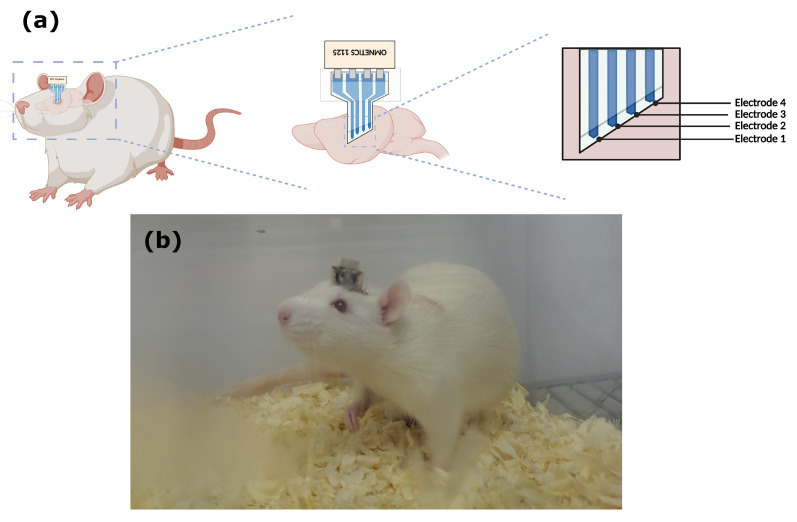
(**a**) Demonstration of the electrode in the rat brain (created with BioRender.com). (**b**) Wistar rat two days after surgery.

**Figure 5 biosensors-12-00853-f005:**
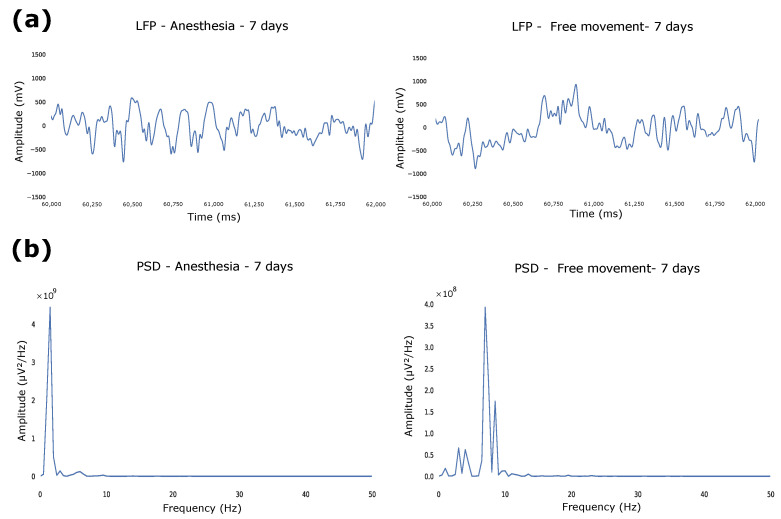
(**a**) LFP between anesthetized and free-movement conditions. (**b**) PSD between anesthetized and free-movement conditions.

**Figure 6 biosensors-12-00853-f006:**
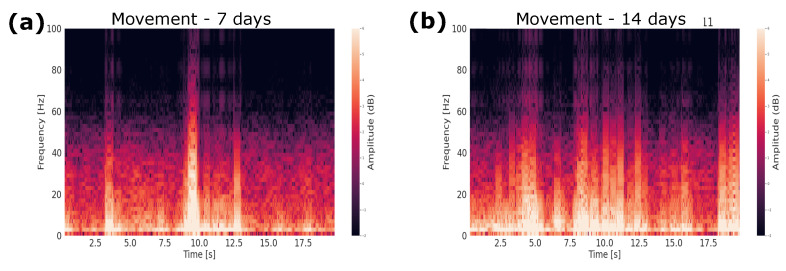
Spectrogram of the moving rat: (**a**) 7 days and (**b**) 14 days.

**Figure 7 biosensors-12-00853-f007:**
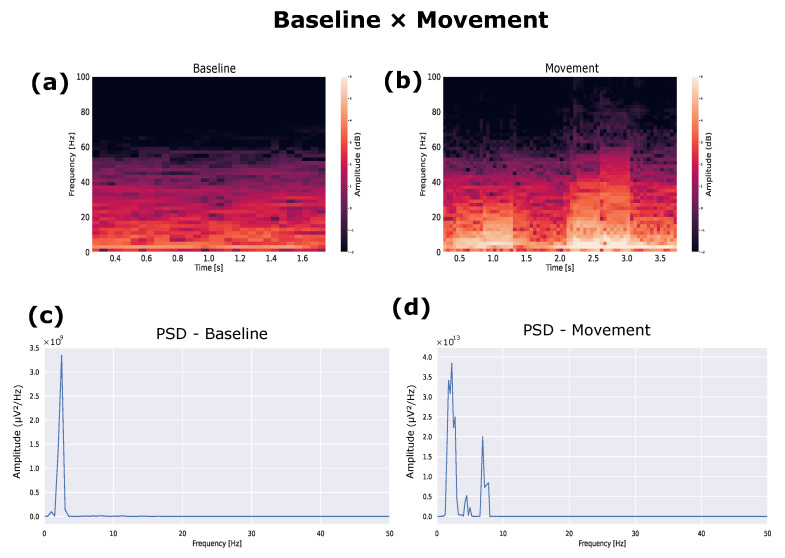
(**a**) Spectrogram of the baseline rat. (**b**) Spectrogram of the moving rat. (**c**) PSD of the baseline rat. (**d**) PSD of the baseline rat during movement.

**Figure 9 biosensors-12-00853-f009:**
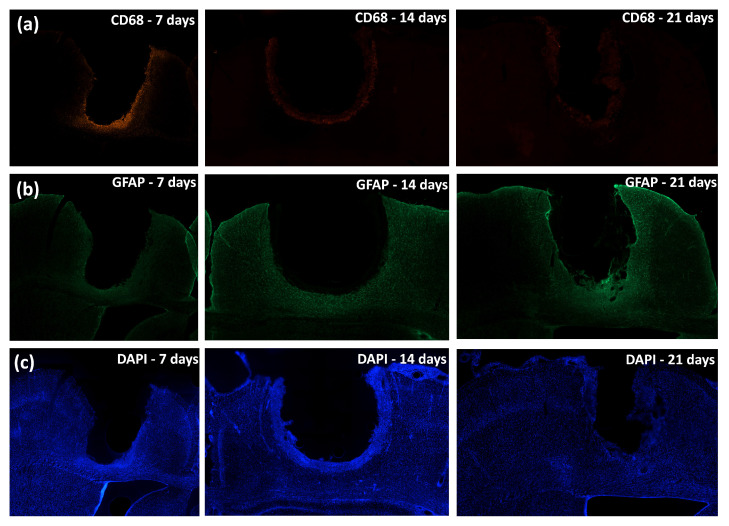
Tissue immune response to implant. (**a**) CD68-labeled cells at days 7, 14, and 21. (**b**) GFAP-labeled cells at days 7, 14, and 21. (**c**) DAPI-labeled cells at days 7, 14, and 21.

**Figure 10 biosensors-12-00853-f010:**
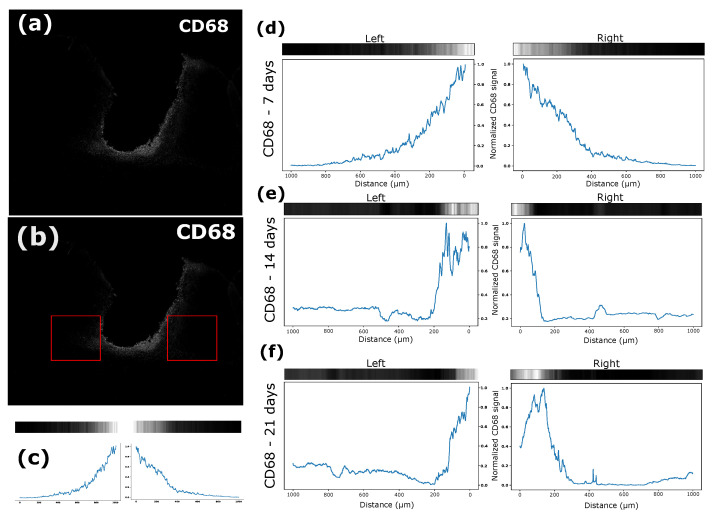
(**a**) Image after treatment with averaging of the intensity vectors. (**b**) Quadrant used to measure CD68 intensity. (**c**) Signal strength near the electrode–tissue interface. (**d**) CD68 signal strength near the electrode–tissue interface after 7 days. (**e**) CD68 signal strength near the electrode–tissue interface after 14 days. (**f**) CD68 signal strength near the electrode–tissue interface after 21 days.

**Figure 12 biosensors-12-00853-f012:**
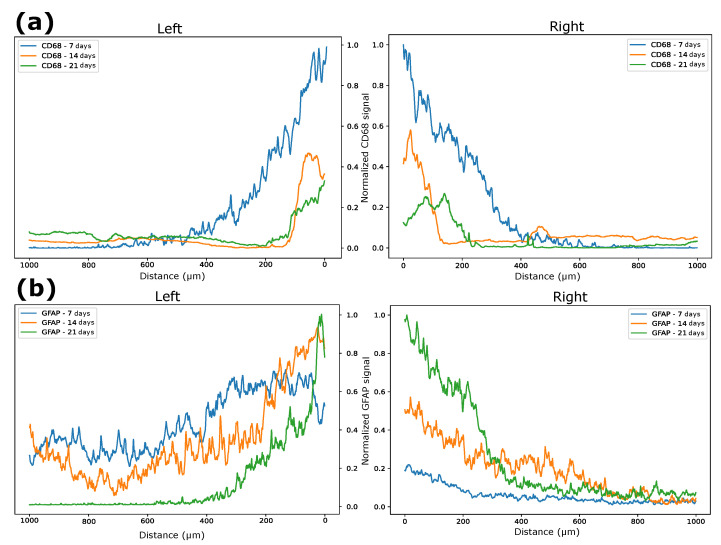
(**a**) Normalized CD68 intensity over 7, 14, and 21 days. (**b**) Normalized GFAP intensity over 7, 14, and 21 days.

## Data Availability

The data that support the findings of this study are available on request from the corresponding author, EM.

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
