# Peer review of "All-Polymeric Electrode Based on PEDOT:PSS for In Vivo Neural Recording"

_biosensors, 2022, doi:10.3390/bios12100853_

Round 1
Reviewer 1 Report
In this manuscript, the authors demonstrate that using all-polymeric PEDOT:PSS electrodes for in-vivo neural recordings. The all-polymeric electrodes exhibit good electrical performance and immune response has limited effect on the in-vivo neural recording property until 21 days. Considering the impact of the work, this reviewer suggests acceptance after minor revision.
Points to be addressed are as follows.
1: The authors utilize a very complicated method to fabricate PEDOT:PSS electrode. First, the author lyophilized the PEDOT:PSS solution and then redispersed it with water/DMSO mixture solvent. However, previous report shows simply adding DMSO to PEDOT:PSS solution is enough to improve electric performance. ( Advanced Electronic Materials 5.3 (2019): 1800804.) Could the author comment why they did not use a simple method? Second, the authors use 3 steps for the device fabrication. (i: 3D printing for the positive mold, ii: cast PDMS on the top for negative PDMS mold, and iii: inject PEDOT:PSS to the PDMS mold manually.) 3D printing is only for the mold fabrication and the electrodes are still deposited manually. Therefore, the author cannot emphasize that they use 3D printing to manufacture electrodes (line 325).
2: Line 38, the authors use the wrong word "notoriety".
3: Line 62 and 64, the values should be 400 µm, 200 µm, and 0.4 µm. I think 0.4 µm PDMS is too thin (it should be 400 µm). Please revise them.
4: Line 197-204, please add the correct units after the number value. e.g. 180.7 ± 19.5 and 7.4 k. Please correct the resting unit issues in the manuscript.
5: Figure format issues. There is no x-axis in Figure 2c. There is no unit in Figure 3. Please use English in Figures 4, 5, and 12. Please add x-axis in Figures 10-11.
Reviewer 2 Report
Filho et al. reported “All-polymeric electrode based on PEDOT:PSS for in-vivo neural recording”. In this work, for neural monitoring in rats, authors created polymeric electrodes based on PEDOT:PSS. They investigated the electrodes' electrical characteristics as well as their in-vitro and in-vivo functioning. They also used histological processing and microscopical inspection to assess the immunological response to the implant after 21 days. They discovered that electrodes with 400-micrometer channels produced a signal-to-noise ratio of 12dB. They reported that in the first 14 days, there was a proliferation of microglia at the tissue-electrode interface, which reduced after 14 days. It was also shown that inflammatory effects had no influence on the signal, indicating that all polymeric electrodes can increase the duration for neural recordings. Overall, the manuscript is well-written. The study seems to comply by the ethical rules and the relevant permissions were obtained by the authors. The data supports the claims of the authors. I have the following comments/questions.
*Authors state that “There was a proliferation of microglia to the tissue-electrode interface in the first days, with a decrease after 14 days” Why microglia proliferation decreases after 14 days? Some explanation could be useful.
* Did the authors consider modification of PDMS surface to improve adhesion?
*Did the authors test the limit of flexibility of the electrodes?
